# Protecting Tear-Film Stability under Adverse Environmental Conditions Using a Mucomimetic with a Non-Newtonian Viscosity Agent

**DOI:** 10.3390/medicina59101862

**Published:** 2023-10-19

**Authors:** Ali Abusharha, E. Ian Pearce, Tayyaba Afsar, Suhail Razak

**Affiliations:** 1Department of Optometry, College of Applied Medical Sciences, King Saud University, Riyadh 11433, Saudi Arabia; aabusharha@ksu.edu.sa; 2Department of Vision Science, School of Health and Life Sciences, Glasgow Caledonian University, 70 Cowcaddence Road, Glasgow G4 0BA, UK; e.i.pearce@gcu.ac.uk; 3Department of Community Health Sciences, College of Applied Medical Sciences, King Saud University, Riyadh 11433, Saudi Arabia; smarazi@ksu.edu.sa

**Keywords:** tamarind-seed polysaccharide (TSP), hyaluronic acid (HA), Rohto Dry Eye Relief drops, drug absorption, ocular surface, teardrops, dry-eye disease

## Abstract

*Background and Objectives:* Tamarind-seed polysaccharide (TSP) and hyaluronic acid (HA) have mucoadhesive properties that improve drug absorption and delay in drug elimination from the ocular surface. We aimed to evaluate TSP/HA-containing formulation for its efficiency in dry-eye symptoms induced by adverse environments and the interaction between mucomimic polymer and tear-film parameters. *Materials and Methods:* The participants were exposed to 5% relative humidity (RH) in a Controlled Environment Chamber (CEC) under constant room temperature (21 °C). Tear-film parameters were assessed at 40% RH and 5% RH. Rohto Dry Eye Relief drops were used in the two treatment modalities, protection (drops instilled before exposure to the dry environment) and relief (drops instilled after exposure to the dry environment). The HIRCAL grid, Servomed EP3 Evaporimeter, and Keeler’s TearScope-Plus were used to screen for non-invasive tear break-up time (NITBUT), tear evaporation rate, and lipid-layer thickness (LLT) using protection and relief treatment methodology. *Results:* LLT was found to be significantly thinner at 5% RH compared with at 40% RH (*p* = 0.007). The median LLT dropped from 50–70 nm (grade 3) at 40% RH to 10–50 nm (grade 2) at 5% RH. TSP/HA eye drops significantly augment LLT in both treatment modalities, protection (*p* = 0.01) and relief (*p* = 0.004) at 5% RH. The mean evaporation rate doubled from 40.93 at 40% RH to 82.42 g/m^2^/h after exposure to 5% RH. In protection mode, the TSP/HA allowed the average evaporation rate to be much lower than when no TSP/HA was used at 5% RH (*p* < 0.008). No alteration in evaporation rate was recorded when the TSP/HA drop was used after exposure (relief). The mean NITBUT was reduced from 13 s in normal conditions to 6 s in the dry environment. Instillation of TSP/HA eye drops resulted in significant improvement (*p* = 0.006) in tear stability, where the NITBUT increased to 8 s in both protection (before exposure) and relief (after exposure) (*p* = 0.001). Although improved, these values were still significantly lower than NITBUT observed at 40% RH. *Conclusions:* Significant protection of tear-film parameters was recorded post instillation of TSP/HA eye drop under a desiccating environment. Both treatment methods (protection and relief) were shown to be effective. The presence of TSP/HA enhances the effectiveness of teardrops in protecting the tear-film parameters when exposed to adverse environments.

## 1. Introduction

Mucoadhesive polymers containing drugs have been used in the management of dry-eye disease (DED) to prolong the residence of ophthalmic solution in the eye [1]. The tear film contains mucin polymers that protect the corneal surface from pathogens and regulate the flow characteristics of the tear film. Recent studies have suggested a correlation between the loss of membrane-linked mucins and premature rupture of the tear film in various eye conditions. Mucin polymers in the tear film protect the corneal surface from pathogens and modulate the tear-film flow characteristics. Recent studies have advocated a relationship between the loss of membrane-linked mucins and premature rupture of the tear film in different eye ailments. The majority of existing mucoadhesive polymers are either polyacrylic acid or cellulose derivatives [2]. Mucomimetic polymers (MMP) can modify both the texture of the aqueous tear and the spreading and structure of the tear-film lipid layer. This enables their synchronized performance in vivo [3].

Tamarind-seed polysaccharide (TSP) is a mucoadhesive polymer extracted from tamarind kernel obtained from tamarind seeds [4,5]. The ability of TSP to enhance ophthalmic-solution delivery and retention in the eye has been documented previously [6,7,8,9]. Similarly, the efficacy of hyaluronic acid (HA) in the management of dry-eye and ocular-surface damage has been well published [10,11,12]. HA stimulates migration, adhesion, and proliferation of corneal epithelium, promoting corneal wound healing [13,14]. HA causes a significant improvement in tear-film stability among patients with meibomian gland dysfunction [11]. In one study, 60 days of treatment with TSP (Xioial, Farmigea, Pisa, Italy) has shown a substantial adjustment in tear stability, ocular comfort, tear production, and reduced ocular-surface epithelial damage [15]. In another study, the use of TSP for 90 days at a concentration of 0.5% and 1% prolonged the non-invasive tear break-up time (NITBUT) significantly in dry-eye patients to 9.60 and 9.40 s, respectively, compared to 5.10 s before the use of TSP. This study also showed that TSP is effective in relieving dry-eye symptoms and corneal damage [16].

Although several studies have investigated mucomimetic polysaccharides, many of these studies focused on the physical parameters of these formulations rather than clinical efficacy. All of these studies have reported that TSP or HA has mucoadhesive proprieties that improve drug absorption and prolong drug elimination from the ocular surface. However, to date, there have been a small number of publications that show how these formulations impact tear-film parameters [17]. Moreover, these studies did not assess the TSP-containing formulation for its efficiency in dry eye induced by adverse environments or the interaction between the mucomimic polymer and tear-film parameters. This type of adverse dry environment was reported to be routinely observed by line crews and manufacturers of high-tech devices where the minimum recorded relative humidity can be less than 5% [18,19,20]. During numerous investigations, the failure to rigorously monitor and control relative humidity (RH) around the eye has been identified as a significant obstacle. As a result, the assessment of the tear film has been compromised. Inaccurate measurements can be attributed to the use of techniques that evaluate the tear film under a range of RH conditions rather than under fixed RH conditions. Therefore, it is imperative to closely monitor and control RH during investigations to ensure the validity and reliability of the study results. The authors could have studied the relationship between tear parameters and the TSP action mechanism for better understanding.

In the current study, we have used an eye formulation (Rohto Dry Eye Drops) that contains a biopolymer of hyaluronic acid and tamarind-seed polysaccharide and is used for the management of dry-eye symptoms. We aim to examine the effectiveness of TSP/HA in protecting tear-film parameters from dry environmental conditions. Moreover, we investigated the relationship between tear-film parameters while using this mucomimetic agent; it is important to evaluate tear-film parameters and their interactions with an ophthalmic solution containing TSP/HA. By exposing normal subjects to a desiccating environment using a control environmental chamber, we have developed a new technique that induces signs and symptoms of dry eye. By doing so, a more comprehensive insight into the effects of TSP on the parameters of tears will be elucidated, contributing to the existing body of knowledge concerning TSP’s therapeutic potential.

## 2. Methods

### 2.1. Controlled Environment Chamber (CEC)

To create requisite environmental conditions, CEC was used. The CEC is an isolated (3 × 3 × 2 m) room planned and constructed by Weiss-Gallenkmap, (Loughborough, UK). The CEC can produce any temperature between 5 °C and 35 °C (±2 °C) and RH between 5% and 95% (±3%). A detailed description of CEC has been provided in our previous lab investigation [21].

### 2.2. Scheme of Investigation

This non-randomized and observational study enrolled twelve healthy male subjects (mean age 29 ± 4 years). All procedures involving human subjects followed ethical standards set forth by the institutional and national committee and adhere to the Helsinki Declaration of 1975, revised in 2019 [22]. Inclusion criteria did not include gender; however, all subjects who participated in this study were male because the recruitment process was done by word of mouth to friends and colleagues. Subjects with a history of ocular surgery, ocular disease, or who wear contact lenses were excluded from this study. Ethical approval was obtained from Glasgow Caledonian University Research Ethics Committee (5 June 2021, GCU1321/34/A). All study procedures were explained to subjects and a consent form was signed by all the subjects. A screening visit was carried out to assess the tear-film and ocular-surface integrity for all participants. Subjects were enrolled in this study if they fulfilled the following inclusion criteria: normal tear stability (NITBUT > 10 s) and a total symptom score of less than 12 from the Ocular Surface Disease Index questionnaire [23].

### 2.3. Treatment Protocol

Rohto Dry Eye Relief drops (Rohto Pharmaceutical Co., Osaka, Japan) were used. Rohto was selected for its novel formulation as it contains a biopolymer of HA and TSP, sodium phosphate dibasic dodecahydrate, mannitol, and sodium phosphate monobasic. The tear film was studied under two different conditions based on RH while the room temperature was constant at 21 °C:
Typical environment (21 °C/40% RH).Dry environment (21 °C/5% RH).

This ultra-dry environmental condition was used to mimic the condition that could be experienced in some artificial buildings or airplane cabins [18,19,20]. All the tear-film investigations were carried out for the right eye. To test protection, dry-eye drops were instilled before exposure to the dry environment. However, in the relief method, the tear supplement was used 15 min post exposure to the dry environment. The design of the protection and relief-testing routine is described in more detail previously [17,24].

### 2.4. Parameters Assessed

For this study, the tear evaporation rate was measured using a Servo-Med Evapometer [25]. Tear break-up time (NITBUT) and lipid-layer thickness (LLT) were evaluated using a Keeler TearScope-Plus and the Guillon and Guillon sorting system [26]. Brief details of the methods used can be found in the section below.

### 2.5. Non-Invasive Tear Break-Up Time (NITBUT)

Tear-film break-up time can be observed non-invasively through observation of the reflected fine-grid image projected on the anterior of the ocular surface and tear film [27]. Throughout this work, the HIRCAL grid and Keeler Tearscope Plus were used to evaluate non-invasive tear break-up time (Figure 1) [21,24]. The HIRCAL is a modified Bausch and Lomb Keratometer with the mire replaced with a white-on-black grid. It has been reported that the NITBUT value of 10 s provides a sensitivity of 82% and specificity of 86% for the diagnosis of DED [28]. A Tearscope Plus (Keeler Ltd., Berkshire, UK) in combination with a non-illuminated biomicroscope was used to observe the quality and thickness of the tear-film lipid layer. The NITBUT was calculated according to the procedure explained by Cho, by taking the average of the three closed NITBUT measurements [29]. The grading system of Guillon and Guillon was used to estimate the classification and, therefore, the LLT. Guillon has suggested that the LLT could be estimated and categorized by its appearance with magnification to five categories, which are open meshwork, closed meshwork, and wave, amorphous and color fringe patterns [26].

### 2.6. Ocular Surface Temperature Measurement (OST)

Using a FLIR System ThermaCAM P620 (FLIR Systems, Surrey, UK), changes in ocular-surface temperature were monitored non-invasively [30]. With a high-definition detector (focal plane array, 640 × 480 pixels), the self-calibrating camera detects temperatures ranging from −40 to +500 °C. A close-up lens with a spatial resolution of 50 µm was attached to the camera to provide a clear and focused image of the ocular surface. Thermal images of the ocular surface were continuously recorded for one minute at a frame rate of 30 Hz. In an Excel spreadsheet, all temperature measurements were exported and 600 thermal measurements were manually selected to exclude the readings taken immediately after blinking (Figure 2) [31].

### 2.7. Tear-Film Evaporation Rate (TFER)

Using a modified Servomed EP3 Evaporimeter (Servo Med, Varberg, Sweden), tear-film evaporation was measured [25]. To accurately determine the evaporation rate, we utilized the highly effective Workbench 5.0 software developed by Strawberry Tree Inc. in Sunnyvale, NS, Canada. This program took 5 readings every second, totaling 600 readings over two minutes. However, to guarantee precise results, we only considered the last 300 readings after allowing ample time for the evaporation values to stabilize. For a more detailed description of this procedure, please refer to our previous data [21].

### 2.8. Statistical Analysis

All the data were analyzed using PASW Statistics version 18 (IBM Corporation, Somers, NY, USA). Firstly, a Kolmogorov–Smirnov test was carried out to check for normality. For the normally distributed data, a repeated measure ANOVA and Tukey’s post hoc test were used for comparison. On the other hand, for the data that were not normally distributed, Friedman’s test was used, followed by a post hoc Wilcoxon rank-sum test. For parameter correlation, a Pearson’s correlation was used for normally distributed data and Spearman’s Rho test for data not following a normal distribution [32]. A level of *p* < 0.05 was considered statistically significant in this study.

## 3. Result

The results obtained for tear-film parameters were analyzed using the comparison of tear parameters measured under the two different environmental conditions and also during both treatment protocols. Table 1 shows the mean and standard deviation of values of tear-film parameters obtained during the assessment of the 12 healthy subjects at 40 and 5% RH, and also when TSP/HA was used for protection and relief during exposure to 5% RH.

### 3.1. Lipid-Layer Thickness (LLT)

LLT was found to be significantly thinner at 5% RH when compared with that observed at 40% RH (*p* = 0.007). The median lipid-layer thickness dropped from 50–70 nm (grade 3) at 40% RH to 10–50 nm (grade 2).

There was a significant increase in LLT following the use of TSP/HA eye drops in both treatment modalities, protection (*p* = 0.01) and relief (*p* = 0.004) at 5% (Table 2). No significant difference was found between protection and relief (*p* = 0.053). The distribution of the lipid-layer patterns observed in normal and desiccating conditions and with the instillation of the TSP/HA is shown in Figure 3.

### 3.2. Evaporation Rate

A significant increase was seen in evaporation rate when the ocular surface was exposed to low RH (*p* < 0.001) (Figure 4). The mean evaporation rate doubled from 40.93 at 40% RH to 82.42 g/m^2^/h 15 min after exposure to 5% RH.

In protection mode, TSP/HA caused the mean evaporation rate to be significantly lower at 56.45 g/m^2^/h (Median = 61.66 g/m^2^/h) compared to when no TSP/HA was used at 5% RH (*p* < 0.008). There was no statistically significant improvement in evaporation when the TSP/HA drop was used after exposure (relief).

### 3.3. Non-Invasive Tear Break-Up Time (NITBUT)

A box plot of tear break-up time assessed at normal (40%), dry condition (5%), and when TSP/HA was used for protection and relief is shown in Figure 5. Tear-film stability was adversely affected by exposure to 5% RH. The mean tear break-up time was reduced from 13 s in normal conditions to 6 s in the dry environment. Instillation of TSP/HA eye drops resulted in significant improvement (*p* = 0.006)) in tear stability, where the NITBUT increased to 8 s in both protection (before exposure) and relief (after exposure) (*p* = 0.001). Although improved, these values were still significantly lower than NITBUT observed at 40% RH. There was no statistically significant difference between both treatment modes (*p* = 0.89).

### 3.4. Correlation Analysis

In this study, a significant correlation was found between tear-film stability and tear evaporation rate (Figure 6). These results emphasize the importance of lipid-layer stability in maintaining tear-film stability prolonging the time of non-invasive tear break-up (NITBUT) and improving tear evaporation rate.

## 4. Discussion

Signs of dry and itchy eyes, nose, and skin are commonly reported in dry conditions [33,34,35]. Attempts have been made to improve indoor working environment [35,36]. Unfortunately, sometimes the surrounding environment cannot be improved, making it difficult to avoid exposure to adverse climate conditions. It is necessary to protect the tear film and ocular surface against these adverse environments. We studied the efficacy of mucomimetic polymers in managing tear-film disruption induced by a desiccating environment using different treatment modalities. Moreover, this study investigated tear-film parameters following perturbation to the tear film caused by low humidity to better understand the interrelationship between the tear-film parameters and the homeostatic mechanism of the tear film. The mucomimetic was tested for two methods of use: protection when it was used before exposure, and relief when it was used after exposure.

Our goal was to check the efficacy of TSP/HA drops in managing the perturbation caused by exposure to dry environments. TSP is a natural hydrophilic mucoadhesive polymer containing glucose, galactose, and xylose monomers, which make up 65% of the polymer, along with soluble and insoluble proteins [37,38]. The use of tamarind gum for the production of the ophthalmic solution was first suggested in 1997 [39]. It has been claimed that TSP could be employed in the production of vehicles for ophthalmic drugs and for use as a tear-film substitute [39]. HA is a linear polysaccharide consisting of D-glucuronic acid and N-acetylglucosamine which belongs to a group of substances known as glycosaminoglycans [40,41]. It is a common component of synovial fluid and the extracellular matrix. In the human body, the highest content of HA is found in the umbilical cord and vitreous humour of the eye [42,43]. A study has shown that TSP 0.7% (*w*/*v*) increased the mean residence time significantly for a variety of ophthalmic solutions when compared to other polymers such as hyaluronic acid (HA), Hydroxyethyl cellulose (HEC), and arabinogalactan [8]. Also, the interaction between rabbit tear mucin and a mixture of different polysaccharides (TSP/HA) has been evaluated [7], and observations indicated that the muco-adhesivity of the TSP/HA mixture is stronger than that of each of the two polymers used alone [7]. HA contributes to tissue hydration and water retention by providing stability and elasticity to the extracellular matrix [44].

A significant improvement in LLT was observed after the instillation of TSP/HA in both treatment protocols. Both the protection and relief techniques successfully improved the thickness of the tear-film lipid layer. The lipid layer plays an important role in controlling tear evaporation and maintains tear-film stability [45,46]. We observed an improvement in tear-film evaporation and stability. This agrees with previous studies suggesting that evaporation and NITBUT improved with a tear supplement containing mucoadhesive polymers such as sodium hyaluronate [47]. TSP is one of the mucomimetic polymers that has been used to thicken and prolong the residence time of ophthalmic solutions [39]. Dry-eye patients have been shown to benefit from HA treatment through increasing tear volume [11]. The improvement in the tear mucous layer, resulting from the use of mucomimetic supplements, leads to the enhancement of tear-film stability, tear spread, and coherence [10,48]. Therefore, the improvement in lipid-layer thickness could be related to improved lipid spreading resulting from the increase in thickness and stability of the underlying aqueous layer.

The evaporation rate of the tear film doubled as a result of exposure to low humidity. It increased sharply from 40.93 to 82.42 g/m^2^/h when the subjects were exposed to 5% relative humidity. This massive increase could be due to the thinning of the lipid layer that was observed. In addition to the thinning of LLT, the increase in evaporation rate could result from the increase in the water-holding capacity at low relative humidity. At low relative humidity, the mass of water vapor in the air is lower, allowing more water molecules to leave the ocular surface and thus increasing the evaporation rate. The elevation in tear evaporation rate may also result from the increase in reflex tearing during exposure to 5% RH. A significant increase in dermal, respiratory airway, and ocular discomfort was observed due to exposure to desiccating environments [49]. Barabino and coworkers elucidated that exposure to a dry environment (RH of 18%) resulted in alternation in tear secretion, corneal staining, and goblet-cell density in animal models [19].

This study suggests that TSP/HA in the protection method was capable of reducing excessive evaporation during dry environments. Water loss from the ocular surface was significantly lower in protection when compared to no drop being used at low humidity (*p* = 0.008). This result was not surprising as the instillation of TSP/HA resulted in a significant increase in tear-lipid-layer thickening. This is in agreement with previous studies, which have shown that the use of Sodium hyaluronate for 90 days reduces tear-film evaporation significantly (*p* = 0.001) in lipid-deficient dry-eye patients [47]. Previous work has shown that hyaluronan polysaccharide helps to delay water loss in vitro due to its sponge-like structure which acts to hold water [44]. Using TSP/HA before exposure (protection) may have allowed more time for the aqueous layer to integrate well with hyaluronan polysaccharide and enhance its ability to control the evaporation rate of tear film. Under dry conditions, HA improves water-retention properties and can reduce evaporation [11].

The use of TSP/HA administrations during exposure to low RH prolonged NITBUT at 5% RH significantly in both treatment methods. Mean NITBUT at 5% RH was 6.24 s which is lower than the typical inter-blink interval [50], and would be classified as grade 2 dry eye according to the DEWS grading scheme of dry eye [51]. This agrees with previous studies suggesting a relationship between lipid-layer stability and tear-film break-up time [52,53]. It was found that all studied tear-film parameters except ocular-surface temperature were adversely affected by exposure to low humidity. A significant increase in evaporation rate, and a reduction in lipid-layer thickness and NITBUT, were recorded when subjects were exposed to a dry environment. Also, as a result of these changes in tear parameters, an increase in dry-eye symptoms were reported more frequently in a desiccating environment. Our findings are in line with various earlier reports mentioning a decrease in NITBUT along with a decrease in RH. Paschides and his colleagues recorded shorter Fluorescein NITBUT (11 s) at 45% compared to that recorded at RH of 55% (15.8 s) [54]. The symptomatic effect of low RH on human-body tissues has been reported in many studies [20,36,55]. In line with our findings, the previous report stated that the mean NITBUT significantly improved at 15, 30 and 60 min time points following the instillation of HA when compared with instances when no HA was used [11]. This correlates well with a study that showed an improvement in NITBUT and a reduction in dry-eye symptoms among dry-eye patients for up to 6 h after using sodium hyaluronate eye drops [12]. Also, it has been reported that contact-lens-care solution containing HA is characterized by lower osmolarity measurement compared to other solutions which may help to enhance ocular-surface and tear-film osmolarity [56]. Similarly, HA has also been found to significantly decrease corneal and conjunctival staining and improve NITBUT in Sjögren syndrome patients when used six times a day for 90 days [10]. Another study reported that TSP improved the NITBUT from 7.1 to 9.6 s among symptomatic contact-lens wearers [15]. Moreover, tear supplements containing 0.5% TSP improved NITBUT in patients with dry eye from 5.15 to 8.45 and 9.64 s after 60 and 90 days of treatment, respectively [16]. It has been shown that mucomimetic polymers can prolong the contact time between eye drops and the ocular surface by adhering to mucin and the epithelial surfaces [9]. Therefore, an improvement in the tear mucous layer leads to enhancement of tear-film stability, tear spread and coherence [48].

## 5. Conclusions

This study’s findings suggest that exposure to a dry environment harmed tear-film parameters. There is a significant improvement of tear-film parameters post instillation of a TSP/HA eye drop when used in a desiccating environment. Both treatment methods were shown to be effective. The presence of TSP/HA enhances the effectiveness of teardrops in managing the key symptoms of dry-eye diseases when exposed to adverse environments. However, it was apparent that using TSP/HA for protection was superior to relief for evaporation rate. Therefore, for maximum benefit, a patient should be advised to use TSP/HA before exposure to dry conditions, such as those found in commercial aircrafts. Moreover, the study makes evident that the use of the CEC could provide researchers with a readily available method for quickly assessing the effectiveness of tear supplementation.

## Figures and Tables

**Figure 1 medicina-59-01862-f001:**
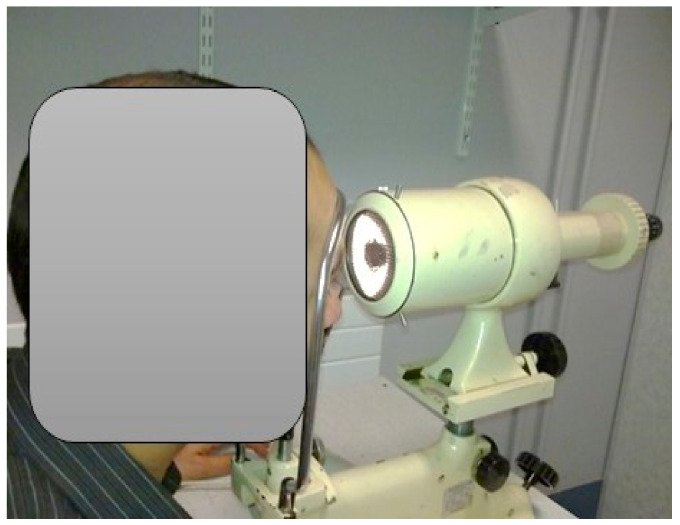
HIRCAL grid. A modified Bausch and Lomb Keratometer fitted with a white-on-black grid. Tear break-up time is estimated by observing the disruption of the projected grid on the tear film.

**Figure 2 medicina-59-01862-f002:**
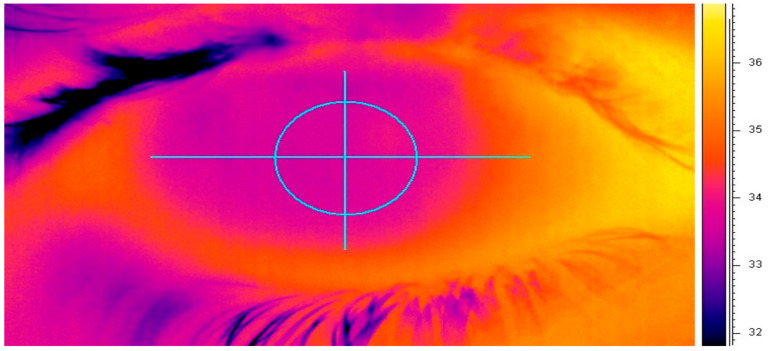
Ocular thermogram displayed on the PC screen using ThemaCAM Researcher software 2.9. The 4 mm circle represents the estimated geometric center of the cornea.

**Figure 3 medicina-59-01862-f003:**
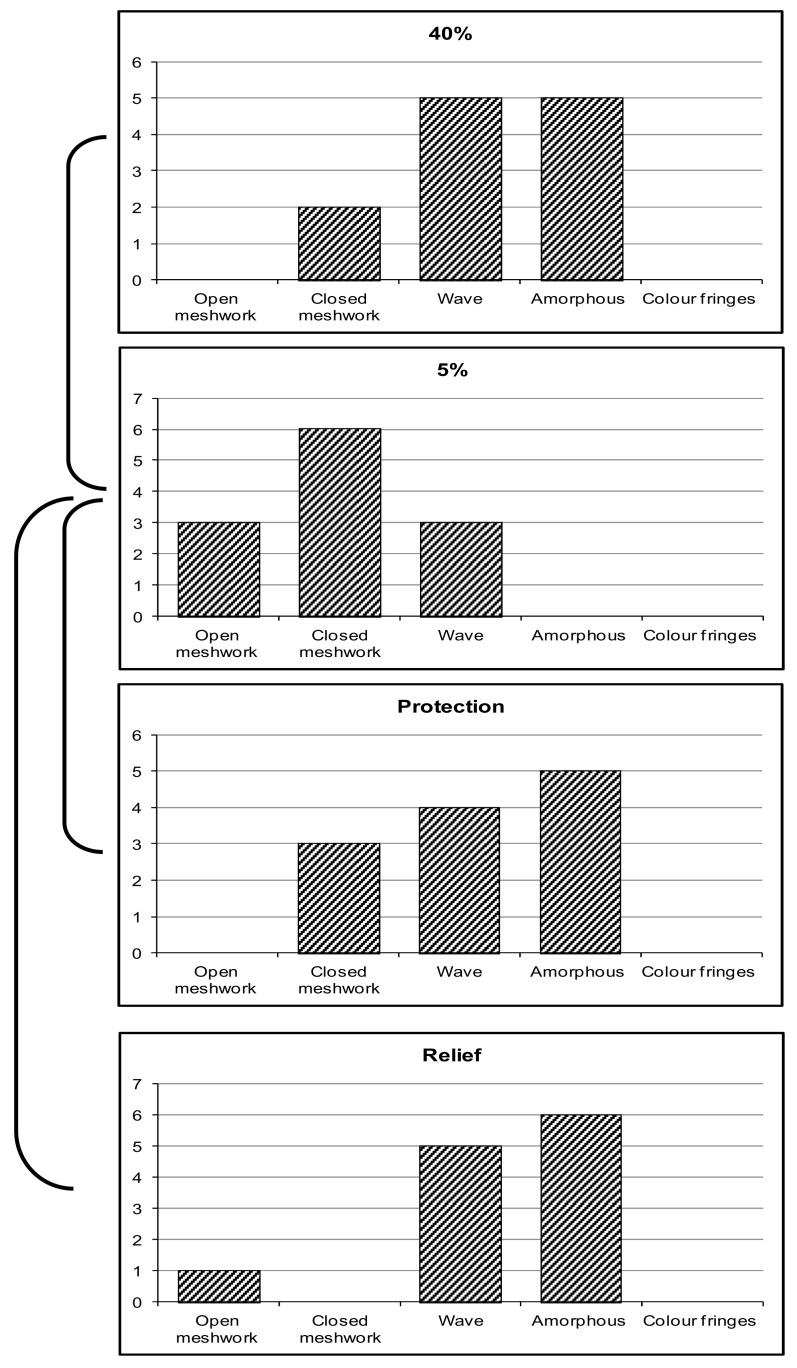
Frequency distribution of lipid-layer-quality grading for 40% and 5% RH, also following the use of TSP/HA for protection and relief. A significant improvement in lipid-layer thickness following the instillation of TSP/HA was observed in both protection and relief.

**Figure 4 medicina-59-01862-f004:**
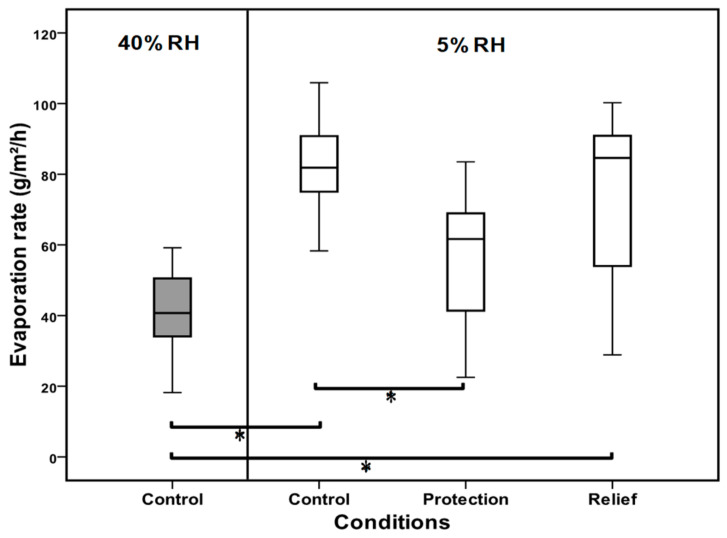
A box plot showing the evaporation rate under normal (40%) and dry (5%) environments, and during the use of TSP/HA before exposure (protection) or following exposure (relief). A significant reduction in evaporation rate was seen post instillation of TSP/HA in protection mode (*p* = 0.008) but not in relief mode (*n* = 12). The box represents the interquartile range that contains 50% of the values. The whiskers are lines that extend from the box to the highest and lowest values. The line across the box indicates the median value. Pairwise, significant differences are indicated by *.

**Figure 5 medicina-59-01862-f005:**
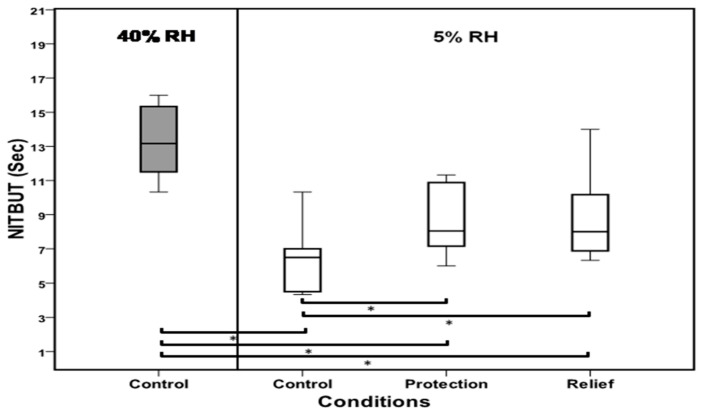
Tear break-up time assessed at 40 and 5% RH, and following the use of TSP/HA-containing drops either before (protection) or after (relief) exposure to 5% RH. A significant reduction in tear stability was observed during exposure to 5% RH. Although NITBUT improved with TSP/HA, these were statically lower than the values at 40% RH (*n* = 12). Both protection and relief improved tear break-up time significantly. The whiskers are lines that extend from the box to the Tear break-up time assessed at 40 and 5% RH, and following the use of TSP/HA-containing drops either before (protection) or after (relief) exposure to 5% RH. The line across the box indicates the median value. Pairwise, significant differences are indicated by *.

**Figure 6 medicina-59-01862-f006:**
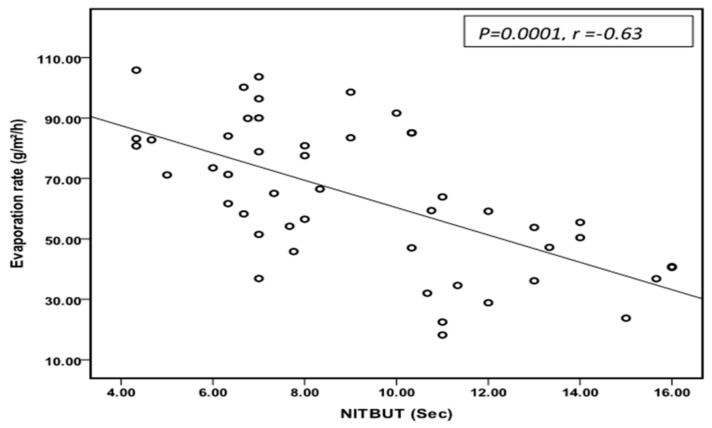
A scatter dot shows the relationship between tear evaporation rate and tear-film stability seen in this study. A significant (*p* = 0.0001) negative correlation was found between these parameters.

**Table 1 medicina-59-01862-t001:** Mean and standard deviation of tear parameters measured at 40 and 5% RH, protection, and relief.

	EVAP (g/m^2^/h)	NITBUT (s)	OST (°C)
**40% RH**	40.93 ± 12.49	13.00 ± 2.1	33.99 ± 0.66
**5% RH**	82.42 ± 14.66	6.00 ± 1.82	33.70 ± 0.44
**Protection**	56.45 ± 18.13	8.00 ± 1.94	33.50 ± 0.41
**Relief**	75.38 ± 22.84	8.00 ± 2.41	33.92 ± 0.59

Evaporation rate (EVAP), non-invasive tear break-up time (NITBUT), and ocular-surface temperature (OST).

**Table 2 medicina-59-01862-t002:** Table of *p* values for LLT differences. Significant differences are underlined.

	**LLT 40%**	LLT 5%	LLT Protection	LLT Relief
**LLT 40%**		0.007	0.725	0.803
**LLT 5%**			0.011	0.004
**LLT Protection**				0.53

## Data Availability

All the relevant data are included in the manuscript. Additional datasets are available upon request from the corresponding author.

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
