# Peer review of "Protecting Tear-Film Stability under Adverse Environmental Conditions Using a Mucomimetic with a Non-Newtonian Viscosity Agent"

_medicina, 2023, doi:10.3390/medicina59101862_

Round 1

Reviewer 1 Report

In the current study, our objective is to investigate how the instillation of artificial tears using various delivery systems can impact tear parameters under both normal environmental conditions and in an environment characterized by low relative humidity. This research topic is highly intriguing, and our data appears to offer valuable insights. Nevertheless, we acknowledge the need for clarification in certain aspects of our methodology. Additionally, we recommend a comprehensive review of the entire document, as we have noticed frequent shifts in writing style and notation that can hinder readability. We suggest conducting a thorough reverse reading of the entire manuscript to ensure consistency. Below, we provide a series of comments in the hope that they will prove useful:

- Line 19: TBUT vs. NITBUT: Please provide clarification; in line 19, TBUT is mentioned, but subsequently, in line 31, there is a reference to NITBUT.

- Lines 19-20: Consider including a brief reference to the conditions employed (as detailed in lines 118-119) to enhance understanding within the abstract.

- Line 26: Please review the use of statistical symbols. There seem to be occasional errors in statistical notation. For instance, in line 26, it may read "p = 0.008" (are you reporting an exact value?). In line 27, "p<0.98" does not seem meaningful. Review the entire notation in both the abstract and the main text.

- Line 28: Regardless of significance, please provide the p-value. It is important to include this information.

- Line 31: Again, a p-value should be provided.

- Line 33: Specify whether you are referring to treatments or modalities.

- Line 34: The conclusion appears to be incorrect as symptoms are not reported to be analysed. Please revise this section.

- Line 40: Consider updating the terminology from "dry eye syndrome" to "dry eye disease" as per Craig et al. (2017, Ocul Surf), both in this section and throughout the manuscript.

- Line 46: "Tear film (TF)" is mentioned here, but it is not consistently used (e.g., 92, 94, 96, 103, 116...). Please standardize its usage.

- Line 62: "Tear break time" seems to differ from "tear break-up time" (line 19). Review the manuscript to ensure consistent notation.

- Lines 46-86: To enhance the relevance of this study, consider adding references to studies on the physical characteristics of tears and artificial tears, particularly their relationship with composition, such as Dalton et al. (2008, OVS) and Pena-Verdeal et al. (2021, CLAE).

- Lines 76-77, 82, 86: Do not use bold for references in these lines.

- Lines 49-86: Consider moving most of this section to the discussion section to improve the flow of the introduction.

- Line 96: The acronym "RH" is used for the first time here, so it should have been introduced earlier (as explained in line 114).

- Lines 106-108: Please relocate this sentence; there should be no sentences or references before stating the study objective. This classification should precede the objective, not follow it.

- Line 111-113: Revise this section for clarity. It should read: "To create the requisite environmental conditions, a controlled environment chamber (CEC) was used. The CEC is an isolated room (3x3x2 meters) planned and constructed by Weiss-Gallenkmap (Loughborough, UK)."

- Lines 116-119: Explain why these specific values of RH were selected. Provide justification.

- Lines 121-134: Clarify the "non-randomized" aspect. Inclusion and exclusion criteria are clearly stated, but the order of the paragraphs is confusing (some details in lines 124-126, others in lines 133-134).

- Lines 120-134: Specify whether both eyes were measured or only one. Reference Armstrong et al. (2013, OPO) and Armstrong et al. (2017, OPO) for clarity.

- Lines 121-123: The enrolment of only 12 subjects may seem like a small sample size. Justify this choice.

- Line 131: This is the first mention of NITBUT, which differs from TBUT as discussed in the abstract. To enhance the scientific value of the manuscript, consider adding a reference in the introduction regarding the relevance of the NITBUT test over the TBUT test, such as Wolffsohn et al. (2017, Ocul Surf).

- Lines 154-164: Indicate how many times NITBUT was measured and specify the type of averaging system used if measured more than once, as recommended due to test variability (Cho et al., 1993, OVS).

- Lines 138-139: Use "HA" and "TSP" instead of "hyaluronic acid and tamarind seed polysaccharide." Review the entire manuscript for such typos to enhance consistency.

- Line 140-141: This information appears to be repeated from a previous section.

- Line 141: The acronym "CEC" has already been introduced.

- Line 142: The acronym "RH" has already been introduced.

- Lines 143-141: This information seems to be reiterated from a previous section.

- Line 161: "DE" is mentioned here, but "dry eye" was more commonly used earlier in the manuscript. Ensure consistency.

- Line 164: Use "LLT" as previously stated (line 151).

- Line 164: Include a brief reference to the various LLT categories, such as the Guillon schema's descriptions, which mainly include Open meshwork, closed meshwork, and wave categories, not just "Grade 1," "Grade 2," or based on nanometre thickness. Clarify the criteria employed.

- Lines 191-197: While the statistical analysis appears appropriate, consider adding references for justification (Armstrong et al., 2011, OPO; Armstrong et al., 2002, OPO).

- Line 201-202: Move the sentence regarding statistical criteria to section 2.8.

- Line 210: Correct the notation to "LLT."

- Figure 3: Review the placement of labels on the axes, as some are difficult to read. Add references to "RH" near "5%" and "40%" for clarity. Consider revising the figure's distribution for better comprehension. The "star" concept is somewhat confusing at a glance; reconsider its presentation.

- Figure 4 vs. Figure 3: These figures seem to have been created by different individuals, with varying notations and legends. Ensure consistency in figure presentation and remove p-values from Figure 4.

- Line 263: Remove the p-value from the table legend or add values to all figures. Consistency in figure legends is crucial for scientific reading fluency.

- Result section notation: Review the notation of p-values. Similar to the abstract, it appears there may be errors in notation (e.g., line 231 should read "p = 0.008").

- Line 305: Correct the typo in ".. we."

- Line 340: Update the reference to more recent sources, such as DESW-II and DEWS lifestyle guidelines, like Craig et al. (2017, Ocul Surf) and Nichols et al. (2023, Ocul Surf).

- Line 355: While we understand the point made, symptoms were not assessed during the treatment process in this study. Therefore, this conclusion cannot be derived from the study design.

Needs revision

Author Response

Comments:

- Line 19: TBUT vs. NITBUT: Please provide clarification; in line 19, TBUT is mentioned, but subsequently, in line 31, there is a reference to NITBUT.

Response: All abbreviations have been changes to NITBUT (non-invasive tear break up time)

- Lines 19-20: Consider including a brief reference to the conditions employed (as detailed in lines 118-119) to enhance understanding within the abstract.

Response: added

- Line 26: Please review the use of statistical symbols. There seem to be occasional errors in statistical notation. For instance, in line 26, it may read "p = 0.008" (are you reporting an exact value?). In line 27, "p<0.98" does not seem meaningful. Review the entire notation in both the abstract and the main text.

Response: Yes, we have reported the exact values. p<0.98 shows no statistical significance but as per your suggestion we have removed this value as the data is not statistically significant so it’s not important to add statistical values.

- Line 28: Regardless of significance, please provide the p-value. It is important to include this information.

Response: added

- Line 31: Again, a p-value should be provided.

Response: added

- Line 33: Specify whether you are referring to treatments or modalities.

Response: Clarified

- Line 34: The conclusion appears to be incorrect as symptoms are not reported to be analysed. Please revise this section.

The conclusion is modified.

- Line 40: Consider updating the terminology from "dry eye syndrome" to "dry eye disease" as per Craig et al. (2017, Ocul Surf), both in this section and throughout the manuscript.

Response: The term has been modified

- Line 46: "Tear film (TF)" is mentioned here, but it is not consistently used (e.g., 92, 94, 96, 103, 116...). Please standardize its usage.

Response: corrected

- Line 62: "Tear break time" seems to differ from "tear break-up time" (line 19). Review the manuscript to ensure consistent notation.

Response: corrected

- Lines 46-86: To enhance the relevance of this study, consider adding references to studies on the physical characteristics of tears and artificial tears, particularly their relationship with composition, such as Dalton et al. (2008, OVS) and Pena-Verdeal et al. (2021, CLAE).

Response: Added

- Lines 76-77, 82, 86: Do not use bold for references in these lines.

Response: Corrected

- Lines 49-86: Consider moving most of this section to the discussion section to improve the flow of the introduction.

Response: introduction modified

- Line 96: The acronym "RH" is used for the first time here, so it should have been introduced earlier (as explained in line 114).

Response: Done

- Lines 106-108: Please relocate this sentence; there should be no sentences or references before stating the study objective. This classification should precede the objective, not follow it.

Response: revised

- Line 111-113: Revise this section for clarity. It should read: "To create the requisite environmental conditions, a controlled environment chamber (CEC) was used. The CEC is an isolated room (3x3x2 meters) planned and constructed by Weiss-Gallenkmap (Loughborough, UK)."

The sentence has been modified as suggested by the reviewer.

- Lines 116-119: Explain why these specific values of RH were selected. Provide justification.

Response: This ultra-dry environmental condition was used to mimic the condition that could be experienced in some artificially building or airplane cabin.

- Lines 121-134: Clarify the "non-randomized" aspect. Inclusion and exclusion criteria are clearly stated, but the order of the paragraphs is confusing (some details in lines 124-126, others in lines 133-134).

All inclusion and exclusion criteria are re-stated again in lines 126 – 128.

- Lines 120-134: Specify whether both eyes were measured or only one. Reference Armstrong et al. (2013, OPO) and Armstrong et al. (2017, OPO) for clarity.

Response: All the tear film investigations were carried out for the right eye during the two treatment modalities (protection and relief)

- Lines 121-123: The enrolment of only 12 subjects may seem like a small sample size. Justify this choice.

Response: We appreciate your thoughtful comments and we agree with your observations and suggestions. Although the sample size is relatively small, many measurements have been conducted. In the current study number of tear film parameters have been evaluated. The human tear film parameters included ocular surface temperature, evaporation rate and non-invasive tear breakup time were evaluated at different time points. Despite that the small sample size may limit the ability to generalize the findings to a larger population, the result of the current study found significant changes in tear film. However, in future investigation it would be useful to evaluate the effect of the tear supplement with a larger sample size that may to increase the statistical power of the analysis.

- Line 131: This is the first mention of NITBUT, which differs from TBUT as discussed in the abstract. To enhance the scientific value of the manuscript, consider adding a reference in the introduction regarding the relevance of the NITBUT test over the TBUT test, such as Wolffsohn et al. (2017, Ocul Surf).

It is modified to non-invasive tear break up time (NITBUT) throughout the manuscript

- Lines 154-164: Indicate how many times NITBUT was measured and specify the type of averaging system used if measured more than once, as recommended due to test variability (Cho et al., 1993, OVS).

Response:  added

- Lines 138-139: Use "HA" and "TSP" instead of "hyaluronic acid and tamarind seed polysaccharide." Review the entire manuscript for such typos to enhance consistency.

Response: corrected

- Line 140-141: This information appears to be repeated from a previous section.

Response: deleted

- Line 141: The acronym "CEC" has already been introduced.

Modified

- Line 142: The acronym "RH" has already been introduced.

Modified

- Lines 143-141: This information seems to be reiterated from a previous section.

Response: deleted

- Line 161: "DE" is mentioned here, but "dry eye" was more commonly used earlier in the manuscript. Ensure consistency.

modified

- Line 164: Use "LLT" as previously stated (line 151).

Modified

- Line 164: Include a brief reference to the various LLT categories, such as the Guillon schema's descriptions, which mainly include Open meshwork, closed meshwork, and wave categories, not just "Grade 1," "Grade 2," or based on nanometre thickness. Clarify the criteria employed.

Response: added as suggested

- Lines 191-197: While the statistical analysis appears appropriate, consider adding references for justification (Armstrong et al., 2011, OPO; Armstrong et al., 2002, OPO).

Response: added

- Line 201-202: Move the sentence regarding statistical criteria to section 2.8.

Response: corrected

- Line 210: Correct the notation to "LLT."

Response: corrected

- Figure 3: Review the placement of labels on the axes, as some are difficult to read. Add references to "RH" near "5%" and "40%" for clarity. Consider revising the figure's distribution for better comprehension. The "star" concept is somewhat confusing at a glance; reconsider its presentation.

Response: revised

- Figure 4 vs. Figure 3: These figures seem to have been created by different individuals, with varying notations and legends. Ensure consistency in figure presentation and remove p-values from Figure 4.

Response: Figures have been modified. The box plot in figure 4 was generated by SPSS while figure 3 was generated by excel due to the difference in figure types and data category. This may explain the difference in appearance between the figures.

- Line 263: Remove the p-value from the table legend or add values to all figures. Consistency in figure legends is crucial for scientific reading fluency.

Response:  corrected as suggested

- Result section notation: Review the notation of p-values. Similar to the abstract, it appears there may be errors in notation (e.g., line 231 should read "p = 0.008").

Response: checked, and added same as in results table. As we measured the tear parameters under two diiferent treatment protocols, it woulb be expected that we found two different p values. The p value in 231 represent the value of the reading measured when the drop was used in relief mode (using the drop after exposure) which is not significant. Whereas the value stated in the abstract is for the protection mode (using the drop before the exposure).

- Line 305: Correct the typo in “we."

Response: corrected

- Line 340: Update the reference to more recent sources, such as DESW-II and DEWS lifestyle guidelines, like Craig et al. (2017, Ocul Surf) and Nichols et al. (2023, Ocul Surf).

Response: added

- Line 355: While we understand the point made, symptoms were not assessed during the treatment process in this study. Therefore, this conclusion cannot be derived from the study design.

The conclusion has been modified and focused more on tear film parameters rather than the symptoms.

Reviewer 2 Report

The authors used relevant professional and scientific literature from this field. It is an original work that brings a contemporary treatment of issues related to the mentioned topic and brings together all the novelties in the literature.  The terminology used in the research is clearly presented. The work is presented transparently and clearly. In terms of language, the work is written legibly and comprehensibly, clearly using English terminology.

Author Response

Dear Reviewer we appreciate your encouragement 

Reviewer 3 Report

Rohto Dry Eye Relief drops were used (for those who are not specialized, you should mention what is Rohoto eye drops about?

adverse environmental conditions using a Con- 17 trolled Environment Chamber (CEC). Highlight these adverse conditions.

prolong drug elimination from the ocular surface. Either replace prolong with delay or elimination with residence to be a meaningful sentence.

Can you provide the LLT for treated groups with TSP/HA eye drops under 5%RH

Replace cellulose byproducts with cellulose derivatives

The objective of this study is two-fold. It is not very common words. Please rearrange the main objectives.

It would be better to include results for individual TSP and HA eye drops separately.

Recent reports indicate the efficacy of trehalose/HA eye drops combination. Why  you have not used such combination for comparative purposes.

Figure 3 is missing error bars.

The manuscript needs to be revised for English

Author Response

Reviewer 3

Rohto Dry Eye Relief drops were used (for those who are not specialized, you should mention what is Rohoto eye drops about?

Response: Rohto dry eye drop is a tear supplement with mucoadhesive polymers that are used for the management of dry eye symptoms. Throughout the manuscript, the role of mucoadhesive polymers and it is mechanism of work have been discussed and reviewed in the introduction (lines 40 – 48). Moreover, the drop composition and its main components (Tamarind seed polysaccharide TSP and hyaluronic acid) have also been discussed and reviewed.  The role of Tamarind seed polysaccharide TSP and hyaluronic acid in managing and controlling dry eye symptoms have also been reviewed in the introduction as well as the discussion section in a way that might help to understand the composition of drops and way of work. However, if more information still needed to be added we are more than happy to do so with consideration of the manuscript length.

adverse environmental conditions using a Con- 17 trolled Environment Chamber (CEC). Highlight these adverse conditions.

Response: As illustrated in the method section the subjects were exposed to ultra-dry environmental conditions with a relative humidity 5%

prolong drug elimination from the ocular surface. Either replace prolong with delay or elimination with residence to be a meaningful sentence.

Response: corrected

Can you provide the LLT for treated groups with TSP/HA eye drops under 5%RH

Response: It is provided in figure 3. Figure 3 shows the distribution of lipid layer quality grading for 40%, 5% RH, also following the use of TSP/HA for protection and relief. The chart marked with Protection and relief represents the LLT following the use of TSP/ HA.

Replace cellulose byproducts with cellulose derivatives

Response: corrected

The objective of this study is two-fold. It is not a very common word. Please rearrange the main objectives.

Response: Revised

It would be better to include results for individual TSP and HA eye drops separately.

Response: We agree with you, however, the composition of Rohto eye drop contains a mixture of both TSP and HA. It would be useful to consider the evaluation of each component separately in future studies to determine which of the two components has the most powerful effect.

Recent reports indicate the efficacy of the trehalose/HA eye drops combination. Why you have not used such a combination for comparative purposes?

Response: Rohto drops are combination of TSP and HA along with other components. Thank you for highlighting this tear supplement formulation. We agree with the reviewer the use of this formulation or other formulations would be very useful for the purpose of the comparison. Numerous tear supplement formulations are taken into consideration in future studies.

Figure 3 is missing error bars.

Figures have been modified.

Round 2

Reviewer 1 Report

No coments

No coments

Reviewer 3 Report

The authors satisfactorily addressed my comments and the quality of the manuscript has been improved. 

None